# Mozambican Adolescents and Youths during the COVID-19 Pandemic: Knowledge and Awareness Gaps in the Provinces of Sofala and Tete

**DOI:** 10.3390/healthcare9030321

**Published:** 2021-03-13

**Authors:** Claudia Marotta, Ussene Nacareia, Alba Sardon Estevez, Francesca Tognon, Giselle Daiana Genna, Giovanna De Meneghi, Edoardo Occa, Lucy Ramirez, Marzia Lazzari, Francesco Di Gennaro, Giovanni Putoto

**Affiliations:** 1Operational Research Unit, Doctors with Africa CUAMM, 35121 Padua, Italy; marotta.claudia@gmail.com (C.M.); f.tognon@cuamm.org (F.T.); g.putoto@cuamm.org (G.P.); 2Dipartimento di Salute della Donna e del Bambino, Università degli Studi di Padova, 35121 Padova, Italy; u.nacareia@cuamm.org (U.N.); a.sardonestevez@cuamm.org (A.S.E.); 3Doctors with Africa CUAMM, Beira 2100, Mozambique; giselledaiana.genna@gmail.com (G.D.G.); g.demeneghi@cuamm.org (G.D.M.); e.occa@cuamm.org (E.O.); l.ramirez@cuamm.org (L.R.); m.lazzari@cuamm.org (M.L.)

**Keywords:** SARS-CoV-2, Mozambique, adolescent, Africa, preparedness, global health

## Abstract

(1) Background: Mozambique has an average population age of 17 years and adolescents and youths have a pivotal role in SARS-CoV-2 pandemic control. (2) Methods: We conducted a cross-sectional study in order to assess the awareness and information needs with regard to COVID-19 among a sample of adolescents and youths from two different Mozambican provinces. (3) Results: Only 25% of adolescents and youths had a high level of awareness and only 543/2170 participants reported a high level of knowledge regarding COVID-19. In our multivariate model, significant predictors of reporting a high level of knowledge about COVID-19 include female sex (O.R. = 1.47; 95% confidence interval (CI) 1.23–2.89), having a house without a thatched roof (O.R. = 1.85; 95% CI 1.02–2.95) and HIV-positive status (O.R. = 1.56; 95% CI 1.36–2.87). (4) Conclusions: Our study highlights an important and relevant knowledge gap in adolescents and youths with respect to the COVID-19 pandemic. Involving young people and adolescents in the fight against SARS-CoV-2 is an essential strategy, especially in countries where the national average age is young, such as Mozambique, and where this epidemic can aggravate an already fragile health system.

## 1. Introduction

Since the first confirmed case of COVID-19 in Algeria on 25 February 2020, more than four million cases of COVID-19 have been reported in the WHO African Region, along with more than 100,000 deaths [1]. Mozambique—one of the last countries to be affected in Africa—reported its first positive case of COVID-19 on 22 March 2020, a 75-year-old Mozambican man who returned from the United Kingdom (UK). Since then, more than 50,000 confirmed cases of COVID-19 have been recorded in Mozambique and Maputo, the capital of the country, has the highest number of cases of SARS-CoV-2 infection, with more 140 deaths [2,3], and 126 districts out of 128 having announced at least one case of COVID-19 [3]. However, data on COVID-19 in Africa and, in particular, in Mozambique, are scarce [4].

The country continues to expand its testing capacity, having carried out less than 300,000 tests since the start of the pandemic so far (roughly equivalent to the number of tests carried out by European countries in two days), which could imply that there is a strong level of under-reporting of COVID-19 cases [5]. 

Mozambique, like most African countries, is primarily made up of young people, with a population that has an average age of 17 years [6]. Even though people of any age can contract COVID-19, the young are less likely than older people to become seriously ill; however, the pandemic is still having a huge impact on the lives of this group. Containment measures, such as lockdowns, the closure of schools and physical distancing, pose many challenges, including interruptions to education and daily routines, increasing levels of domestic violence, stress and mental health issues [7]. Moreover, the role of young people is well established as pivotal to pandemic control, due to the fact that this age group has a higher number of asymptomatic cases, thereby acting as potential vectors of SARS-CoV-2 infection within communities and families [8]. However, if young people are empowered, inspired, engaged, and given the opportunity to lead, they will rise to these challenges, help create community resilience, and drive social change during the pandemic. Furthermore, young people are more frequently exposed to social media and television than other age groups.

Among the important aspects when referring to Mozambique and to the complexity of the effects that the pandemic could have in this context are the high levels of endemic HIV, tuberculosis and malaria in the country. On the one hand, the presence of these diseases reflects the presence of risk factors for a worse outcome following SARS-CoV-2 infection and, on the other hand, highlights how the response to the COVID-19 pandemic can shift and drastically reduce the economic and human resources provided for the battle against these major killers (HIV, TB and malaria) [9,10,11,12]. Moreover, especially for HIV, as has already occurred in other cases in the past [13], this may also lead to a decline in HIV care, with a risk of lower levels of clinical follow-up, a challenge in obtaining antiretroviral medications, clinical worsening in the short and long term and, more generally, a reduction in disease control [14]. 

The purpose of our research is therefore to study awareness and understanding of SARS-CoV-2 in young, HIV-positive people from two provinces in Mozambique: Sofala and Tete. The knowledge gap we expect to document and identify could be helpful in informing targeted interventions.

## 2. Materials and Methods

### 2.1. Study Setting, Design and Population

Since 1978, Doctors with Africa—CUAMM have had the goal of enhancing the provision of specific facilities for adolescents and young people in Mozambique (10–24 years), from creating healthcare units, *Servico amigo do adolescente e jovem* (SAAJ), schools and communities, building relationships between these services, and providing technical assistance to the health authorities of the Provincial Directorate of Health of Tete and Sofala.

As part of the COVID-19 emergency response, between April and July 2020, Doctors with Afr—CUAMM conducted a survey study using a cross-sectional methodology [15] in order to intercept the awareness and information needs regarding COVID-19 among a sample of adolescents and youths attending 10 SAAJs supported by Doctors with Afr—CUAMM:Three in the Province of Tete, one for each of the following districts: Angonia, Moatize and Tete City;Seven in the Province of Sofala, all in the district of Beira City, where only adolescents and youths living with HIV were considered.

The eligible population included all persons aged 10–24 years who were enrolled on the lists of the SAAJ.

Exclusion criteria were the unavailability of a telephone number contained within the SAAJ registry, no response after at least three attempts (two attempts in the morning/afternoon and one attempt in the evening), refusal to answer the questionnaire, and institutionalization (convent, prison, hospital, etc.).

A standardized questionnaire was administered through a telephone interview by healthcare activists. The telephone numbers of participants were collected through the patient lists of adolescent services (SAAJs) within the health centers.

Training was provided to all interviewers to standardize the procedures and to ensure the quality of data collection. At the beginning of the interview, informed consent was obtained and the survey aims were explained, as well as the methods used to ensure the confidentiality of the data. At the end of the interview, participants received health advice if requested.

The collected data were entered in a dedicated database and a quality control check of the data entry was performed before data analysis.

The questionnaire was made up of questions with multiple and open-text answers, divided into four sections: (I) socio-demographic information, (II) knowledge of COVID-19 signs and symptoms, (III) knowledge of preventive measures and risk factors, and (IV), for HIV patients, a final section, consisting of questions on the challenges of antiretroviral treatment in the context of the COVID-19 pandemic.

### 2.2. Statistical Analysis

Descriptive analysis was performed to define the distribution of the characteristics of the sample and a χ2 test (with Fisher’s correction if less than five cases were present in a cell) was applied for categorical variables. The study outcome was having a “high level of knowledge on COVID-19”. The role of HIV status against the level of knowledge was also investigated within the analysis.

Participants’ level of knowledge with regard to COVID-19 was classified as low–medium and high on the basis of answers provided to the questions about COVID-19 signs and symptoms, risk factors/transmission and prevention measures. A logistic regression model was implemented as follows. A high level of knowledge on COVID-19 was considered as a dependent variable and each one of the available factors at the baseline evaluation as independent variables (univariate analysis). In the multivariate analysis, factors with a *p*-value < 0.10, as assessed by univariate analysis, were included. Multicollinearity among covariates was assessed through the variance inflation factor, taking a value of 2 as cut-off to exclude a covariate. However, no variables were excluded according to this pre-specified criterion. Odds ratios (O.R.s) as adjusted odds ratios (adj-O.R.s) with 95% confidence intervals (CIs) were used to measure the strength of the association between factors at the baseline (exposure) and high level of knowledge about COVID-19 (outcome). All statistical tests were two-tailed and statistical significance was assumed for a *p*-value < 0.05. Statistical analyses were performed with GraphPad Prism version 8.0 (GraphPad Software, Inc., San Diego, CA, USA).

## 3. Results

Between April and July, a total of 2170 Mozambiquan adolescents and youths agreed to participate in the study and were interviewed: 1580 (F 964, 61%) were from Tete and 590 (F 445, 75%) were from Sofala province. Demographic and socio-economic characteristics, compared by province, are shown in Table 1.

On the basis of the answers provided to questions regarding signs and symptoms, risk factors/transmission and prevention measures for SARS-CoV-2 infection, participants’ awareness was measured.

Since participants from Sofala Province were all HIV positive, extra questions about difficulties in antiretroviral treatment supply during COVID-19 were provided. While 99.7% of them reported that they had received antiretroviral drugs during the pandemic, 90% were not able to estimate the length in days of the coverage provided by the medicine supplied to them (Table 2).

A total of 543 (25%) participants reported a high level of knowledge about COVID-19.

No difference was documented between participants reporting different levels of knowledge as regards having a house with a sand floor (*p*-value: 0.078) and awareness about the COVID-19 pandemic (*p*-value: 0.12). On the contrary, older age (age group 20–24 y 75%; *p*-value: 0.006), being male (79%, *p*-value: 0.001), living in a house without a water supply (93%, *p*-value: <0.0001), without a latrine (94%, *p*-value: <0.0001), with a thatched roof (91%, *p*-value: <0.0001) and with the availability of soap and water for hand washing (86%, *p*-value: <0.0001) were more frequent in participants with a low–medium level of knowledge of COVID-19 (Table 3).

The multivariate model considered the effects of age, HIV status, living in a house without a latrine, with a thatched roof, with the availability of soap and water for hand washing and different sources of health information. Significant predictors of reporting a high level of knowledge of COVID-19, reported in Table 4, include: female sex (O.R. = 1.47; 95% CI 1.23–2.89), having a house without a thatched roof (O.R. = 1.85; 95% CI 1.02–2.95) and HIV-positive status (O.R. = 1.56; 95% CI 1.36–2.87).

## 4. Discussion

Our study evaluates the knowledge and awareness gaps with regard to SARS-CoV-2 in 2170 Mozambican adolescents and youths from two different provinces (Sofala and Tete), and four districts (Tete City, Angonia, Moatize, Beira City). Despite the fact that the pandemic has been ongoing for a year at this point, there was still a huge knowledge gap. In fact, only 25% of adolescents and youths had a high level of awareness and, similarly, only 25% of participants reported a high level of knowledge about COVID-19. Older age, being male, living in a house without a water supply, without a latrine, with a thatched roof and with the availability of soap and water for hand washing were more frequent in participants reporting a low–medium level of knowledge about COVID-19. In the multivariate model, significant predictors of a high level of knowledge about COVID-19 were being female, having a house without a thatched roof and having a HIV-positive status. Furthermore, 72% of respondents reported that they seek information about SARS-CoV-2 from the media. As for the sub-sample, made up of HIV-positive adolescents and youths (590/2170), when asked about the coverage of drugs, 90% of them declared that they did not remember the quantity of medication available to them in order to avoid a possible lack of a follow-up visit due to the pandemic.

Few data are available on the knowledge of COVID-19 among adolescents and youths, especially in African countries [16]. At-risk populations’ demographic characteristics play a vital role in the type and intensity of measures necessary to curb the spread of the virus. Improving the knowledge of an infectious disease is usually considered the first approach to any implementable health mitigation strategy, including increasing public awareness of preventive measures to stop transmission [17]. For these reasons, adolescents can be an important resource in mitigating strategies and community outreach in a pandemic, especially in the African context. As several authors report, SARS-CoV-2 infection in adolescents is more often asymptomatic and/or paucisymptomatic, but they play a central role in the spread of the virus, as they are vehicles for infection within communities and families [18,19]. Furthermore, African countries—especially Mozambique—are primarily made up of young people. In fact, considering that Mozambique has an average age of 17 years, investigating the level of knowledge of young people with regard to COVID-19 allows us to indirectly assess the level of awareness of the pandemic in the general population as a whole. Our data are similar to a study carried out in the South West Region of Cameroon, where almost 50% of the study population showed a low level of knowledge about COVID-19, while a study from Nigeria showed a higher level of knowledge. The authors correlate this difference with the composition of the study sample, which was made up of participants of a high socio-economic level [20,21]. Furthermore, in our study, in both provinces investigated, adolescents and young people claimed to receive information on SARS-CoV-2 from the media, such as the radio, TV and Internet. As other studies from the US and China have underlined, the role of social media can be twofold: on the one hand, it allows information to be disseminated to a large extent, but, on the other hand, information from these sources are not verified and there is a high risk of misinformation, which can negatively impact any mitigation strategies against the pandemic [22,23,24]. Therefore, it is recommended that specific programs be developed to target young people, with effective communication strategies with regard to preventative measures against COVID-19, in order to allow for greater dissemination and adherence within communities. In our multivariate analysis, a high level of knowledge was associated with being female and HIV-positive status. With regard to HIV status, this could be explained by the fact that HIV-positive adolescents are targeted by programs with a focus on the importance of behaviors, health measures, and therapy adherence. Moreover, community activists play a huge role in guarantying health education and awareness within the community in order to fight against the lost-to-follow-up phenomenon [13,25]. For these reasons, it could be hypothesized that, for this population, keeping in contact with a health personnel, both through outpatient services and/or outreach initiatives by health activists, gave them the chance to be informed about COVID-19 as well as HIV.

Attention should be paid to the findings regarding medicine coverage, since many HIV-positive youths declared that they did not have enough medicine, or could not remember the exact timeframe of the coverage provided by the medicine they had. This can be difficult to interpret, as the number of patients lost to follow-up is high in Mozambique, and we do not know how much the COVID-19 pandemic has affected the interruption of HIV services and therapeutic continuity, which can have devastating effects due to the worsening of the clinical stage and resistance to medium-term therapeutic treatments and drugs. Moreover, several sources have shown that the COVID-19 outbreak is accelerating, with a transition to widespread community transmission [26,27,28].

We recognize that there are some limitations to our study: first of all, we presented data from a convenience sample (a non-representative sample), related only to two provinces in Mozambique, and unbalanced for variables such as age and gender. This should be considered when generalizing these results, as their external validity could be weak. On the contrary, this data collection represents an important struggle when considering the complexity of the rural settings studied. Moreover, more variables on factors that potentially influence knowledge and awareness, such as socioeconomic characteristics, e.g., income, educational status and evaluation of personal experiences of COVID-19, as well as the role of schools in health education, could have been collected and considered in the analysis.

Further research, implementing other study designs focusing on the determinants that emerged through our preliminary descriptive analysis, should be considered.

## 5. Conclusions

In conclusion, our study outlines an important and relevant knowledge gap in a sample of Mozambican adolescents and youths with respect to the COVID-19 pandemic. An urgent need to reinforce education on preventive measures and intensify sensitization campaigns has emerged. The active engagement of young people and adolescents in the fight against SARS-CoV-2 could be an essential strategy, especially in countries where the national average age is very low, such as Mozambique, and where the pandemic can aggravate the already fragile health system, in a country that is already struggling with “chronic epidemics”, such as those of malaria, HIV and tuberculosis.

Interestingly, the WHO’s Regional Office for the Eastern Mediterranean produced a youth engagement framework, Youth for Health, which identifies three key objectives in any initiative intended to engage young people: empowerment, action and [29] participation. In the same direction, field experiences with regard to adolescent and youth engagement, reported in other countries such as Guinea Bissau [30], could be widely implemented after adapting them to the present context.

It is also recommended that public policies to reduce misinformation and misunderstandings about COVID-19 are designed, alongside the creation of communication strategies to avoid stigma or discrimination against people living with HIV/AIDS and/or those who have contracted COVID-19 in order to prevent people who suffer from these diseases from presenting themselves, without prejudice, to health facilities for proper monitoring and treatment. Moreover, maintaining outpatient services for HIV patients to guarantee the continuity of antiretroviral treatments and avoid devastating effects on HIV control in the short, medium and long term is crucial.

## Figures and Tables

**Table 1 healthcare-09-00321-t001:** Demographic and socio-economic characteristics and level of knowledge about COVID-19 signs and symptoms, risk factors/transmission and prevention measures, compared by participants being HIV positive (Sofala province) or not (Tete).

			Total n. 2170 n (%)	Province of Tete n. 1580 n (%)	Province of Sofala n. 590 n (%)
Demographic information	Sex	M	761 (35)	616 (39)	145 (25)
		F	1409 (65)	964 (61)	445 (75)
	Age	10–14 y	91 (4)	6 (0.4)	85 (14)
		15–19 y	145 (7)	0 (0)	145 (25)
		20–24 y	1934 (89)	1574 (99.6%)	360 (61%)
	District	Tete City	429 (20)	429 (27)	-
		Angonia	752 (35)	752 (48)	-
		Moatize	399 (18)	399 (25)	-
		Beira City	590 (27)	0 (0)	590 (100)
	HIV status	positive	590 (100)	-	590 (100)
Socio-economic factors	Number of family members, median		4.9	4.8	5.1
	House without water supply		548 (25)	490 (31)	58 (10)
	House without latrine		170(78)	1339 (85)	365 (62)
	Thatched roof		385 (18)	372 (24)	13 (2)
	Sand floor/non-washable floor		501 (23)	433 (28)	68 (12)
	Availability of soap and water for hand washing		1752 (81)	1182 (81)	570 (97)
	Awareness about COVID-19 pandemic		2155(99)	1570 (99)	585 (99)
Level of knowledge	Signs and symptoms	Low	858(39)	711 (45)	147 (25)
		Medium	898(41)	632 (40)	266 (45)
		High	544(20)	237 (15)	307 (20)
	Risk factors/transmission	Low	747(35)	600 (38)	147 (25)
		Medium	878(40)	648 (41)	230 (39)
		High	545(25)	332 (21)	213 (36)
	Prevention measures	Low	740(34)	600 (38)	140 (24)
		Medium	829(38)	660 (42)	229 (40)
		High	541(25)	320 (20)	221 (37)
	Source of information on COVID-19	Health center	505(23)	415 (26)	90 (15)
		Media (TV/radio/Internet/social media)	1559(72)	1072 (68)	487 (83)
		Other people	105(5)	93 (6)	13 (2)

**Table 2 healthcare-09-00321-t002:** Antiretroviral treatment during COVID-19 pandemic.

		Sofala Provinces (n. 590)
Received antiretroviral drugs during the pandemic	Yes	588 (99.7%)
	No	2 (0.3%)
Coverage of medicines (days)	30 days	25 (4%)
	60 days	1 (0%)
	90 days	34% (6%)
	Not known	530 (90%)

**Table 3 healthcare-09-00321-t003:** Comparison between levels of knowledge reported by participants.

				Level of Knowledge		*p*-Value
			Total n. 2170 n (%)	High n.543 n (%)	Low–Medium n.1627 n (%)	
Demographic information	Sex	M	761 (35)	159(21)	602(79)	0.001
		F	1409 (65)	384(27)	1025(73)	
	Age	10–14 y	91 (4)	10 (10)	81(90)	0.006
		15–19 y	145 (7)	39(27)	106 (73)	
		20–24 y	1934 (89)	494(25)	1440 (75)	
	HIV status	positive	590 (100)	328 (56)	262 (44)	<0.0001
Socio-economic factors	Number of family members, median		4.9	4.9	5.1	-
	House without water supply		548 (25)	41(7)	507(93)	<0.0001
	House without latrine		1704 (78)	98(6)	1606(94)	<0.0001
	House with thatched roof		385 (18)	43(9)	342(91)	<0.0001
	House with sand floor/non-washable floor		501 (23)	110(22)	391(78)	0.078
	Availability of soap and water for hand washing at home		1752 (81)	251 (14)	1501(86)	<0.0001
	Awareness about COVID-19 pandemic		2155(99)	562 (26)	1593 (74)	0,12
	Source of information on COVID-19	Health center	506(23)	198 (39)	308 (61)	<0.0001
		Media (TV/radio/Internet/social media)	1559(72)	342(22)	1217 (78)	
		Other people	105(5)	4 (4)	101 (96)	

**Table 4 healthcare-09-00321-t004:** Predictors of high level of knowledge about COVID-19.

		OR	95% CI	*p*-Value
Age		0.986	0.912–1.624	0.287
Sex	Male	ref	1.23–2.89	0.003
	Female	1.47		
HIV status	Negative	ref	1.36–2.87	0.010
	Positive	1.56		
House without latrine	No	ref	0.85–2.37	0.168
	Yes	1.03		
House without thatched roof	No	ref	1.02–2.95	0.001
	Yes	1.85		
Availability of soap and water for hand Washing at home	No	ref	0.27–2.14	0.776
	Yes	0.85		
Source of health information	Media	ref	0.61–2.10	0.230
	Health center	1.34		

## Data Availability

Not applicable.

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
