# Peer review of "Mozambican Adolescents and Youths during the COVID-19 Pandemic: Knowledge and Awareness Gaps in the Provinces of Sofala and Tete"

_healthcare, 2021, doi:10.3390/healthcare9030321_

Round 1

Reviewer 1 Report

This work outlines knowledge gaps in adolescent and youth with respect to the COVID-19 pandemic.

The topic is interesting but the whole paper lacks of details. I miss a Background section providing some description on the topic or similar studies. Also, limitations of the study are not discussed. 

In general, the results need to be further discussed as well as their implications.

Author Response

To Healtcare Editor and reviewers,

We have appreciated the positive feedback to our manuscript “Mozambican Adolescents and Youths in COVID-19 Pandemic: knowledge and awareness gaps in the provinces of Sofala and Tete”.

We have considered all the useful suggestions made by the referees and we have implemented the text. We have also satisfied the technical requirements according to the journal guidelines. Modifications have been highlighted using "track changes" feature. Also, a native English speaker has been engaged to improve the fluency and the readability of the manuscript.

We believe that the revision proposed by the referees, and further implemented in the text, contributed to improve the manuscript. Thus, we kindly ask to and re-consider the manuscript for publication.

Please find a point-by-point response to the referees’ comments below.

Best regards,

Dr. Francesco Di Gennaro

Reviewer's Responses to Questions

Reviewer 1

This work outlines knowledge gaps in adolescent and youth with respect to the COVID-19 pandemic.

The topic is interesting but the whole paper lacks of details. I miss a Background section providing some description on the topic or similar studies. Also, limitations of the study are not discussed. 

In general, the results need to be further discussed as well as their implications.

Response

We thank you very much for the encouraging feedback on our manuscript.  We have considered all the useful suggestions to improve our research.

In particular, both the introduction and discussion section are been deeply revised according with your suggestion.

Reviewer 2 Report

Dear authors,

The paper is good but there are some improvements that could be necessary:

1) Regarding methodology and data collection, i think you could present more in detail the questionnaire you administered by telephone, providing some questions examples, answering structure( free, multiple choices,likert?) and scoring method underlining how informations  and awareness about Covid-19 have been operazionalyzed.

2) The sample is really unbalanced for sample size  between two provinces (participants from Tete are more than twice the Sofala's ones)  for gender( most are females, especially from Sofala province) and for age (almost all participants from Tete are aged 20-24) these issues about sample representativeness could be presented more specifically among "limitations".

Author Response

To Healtcare Editor and reviewers,

We have appreciated the positive feedback to our manuscript “Mozambican Adolescents and Youths in COVID-19 Pandemic: knowledge and awareness gaps in the provinces of Sofala and Tete”.

We have considered all the useful suggestions made by the referees and we have implemented the text. We have also satisfied the technical requirements according to the journal guidelines. Modifications have been highlighted using "track changes" feature. Also, a native English speaker has been engaged to improve the fluency and the readability of the manuscript.

We believe that the revision proposed by the referees, and further implemented in the text, contributed to improve the manuscript. Thus, we kindly ask to and re-consider the manuscript for publication.

Please find a point-by-point response to the referees’ comments below.

Best regards,

Dr. Francesco Di Gennaro

Reviewer 2

the paper is good but there are some improvements that could be necessary:

1) Regarding methodology and data collection, i think you could present more in detail the questionnaire you administered by telephone, providing some questions examples, answering structure (free, multiple choices,likert?).

2) The sample is really unbalanced for sample size between two provinces (participants from Tete are more than twice the Sofala's ones)  for gender( most are females, especially from Sofala province) and for age (almost all participants from Tete are aged 20-24) these issues about sample representativeness could be presented more specifically among "limitations".

Response:

We thank you very much for your feedback on our manuscript.  We have considered all the useful suggestions to improve our research. Below point by point response. Furthermore, A native English speaker has been engaged to improve the fluency and the readability of the manuscript.

  1. Regarding methodology and data collection we added all the useful information about the interview and the questionnaire as follow:

A standardized questionnaire was administered through a telephone interview by healthcare activists. [..]

The questionnaire was made up in questions with multiple and open text answers, divided into four sections: (I) socio-demographic information, (II) knowledge on COVID-19 signs and symptoms, (III) knowledge on preventive measures and risk factors, and (IV) for HIV patients a final part consisting of questions on challenges on antiretroviral treatment in the context of the COVID-19 pandemic was added.”

  1. Welcoming your suggestions we discussed the limitations of the sample composition in the discussion section as follow

We recognize some limitations in our study: first of all we presented data from a convenience sample related only to two provinces of Mozambique and unbalanced for variables such us age and gender. This should be considered when generalizing these results as their external validity could be weak. On the contrary this data collection represents an important struggle considered the complexity of the rural settings considered.”

Reviewer 3 Report

The manuscript entitled “Mozambican Adolescents and Youths in COVID-19 Pandemic: knowledge and awareness gaps in the provinces of Sofala and Tete” by Claudia Marotta et al. is significant focusing on need to reinforce measures to fill the knowledge gap in youth of Mozambique with regards to COVID-19.

The methods is presented properly. Though the preventive measures and sensitization campaigns that the authors proposed should be described in more details

I recommend that this manuscript can be published after minor revision.

Author Response

To Healtcare Editor and reviewers,

We have appreciated the positive feedback to our manuscript “Mozambican Adolescents and Youths in COVID-19 Pandemic: knowledge and awareness gaps in the provinces of Sofala and Tete”.

We have considered all the useful suggestions made by the referees and we have implemented the text. We have also satisfied the technical requirements according to the journal guidelines. Modifications have been highlighted using "track changes" feature. Also, a native English speaker has been engaged to improve the fluency and the readability of the manuscript.

We believe that the revision proposed by the referees, and further implemented in the text, contributed to improve the manuscript. Thus, we kindly ask to and re-consider the manuscript for publication.

Please find a point-by-point response to the referees’ comments below.

Best regards,

Dr. Francesco Di Gennaro

Reviewer 3

The manuscript entitled “Mozambican Adolescents and Youths in COVID-19 Pandemic: knowledge and awareness gaps in the provinces of Sofala and Tete” by Claudia Marotta et al. is significant focusing on need to reinforce measures to fill the knowledge gap in youth of Mozambique with regards to COVID-19.

The methods is presented properly. Though the preventive measures and sensitization campaigns that the authors proposed should be described in more details.

I recommend that this manuscript can be published after minor revision.

Response:

We thank you very much for the encouraging feedback on our manuscript.  We have considered all the useful suggestions to improve our research also extending the bibliography with best practices in this fields:

“Of interest WHO’s Regional Office for the Eastern Mediterranean has produced a youth engagement framework, Youth for Health, which identifies three key objectives in any initiative to engage young people: empowerment, action and participation http://www.emro.who.int/pdf/media/news/engaging-young-people-in-the-response-to-covid-19-in-whos-eastern-mediterranean-region.pdf?ua=1, and in the same direction fields experiences adolescent and youth engagement, reported in other countries such as Guinea Bissau  https://www.unv.org/Success-stories/Engaging-women-and-youth-fighting-COVID-19, could be widely implemented after adapting them to the context

Reviewer 4 Report

The study is interesting and strategically important because it adopts a positive process based on information that should drive the most appropriate health planning. Such an approach is not easy and frequent in LMICs such as Mozambique. It must also be promoted by supporting this kind of study through a careful and proper peer review.

In conclusion, this study should be published after major revision  

General remarks.

It should be emphasized that this study is a preliminary descriptive one that will be developed by implementing other study designs focusing on the determinants you investigated so far.

It should be paid attention to the starting research hypothesis, which should be defined at the beginning of the article

Specific remarks

See enclosed

Author Response

To Healtcare Editor and reviewers,

We have appreciated the positive feedback to our manuscript “Mozambican Adolescents and Youths in COVID-19 Pandemic: knowledge and awareness gaps in the provinces of Sofala and Tete”.

We have considered all the useful suggestions made by the referees and we have implemented the text. We have also satisfied the technical requirements according to the journal guidelines. Modifications have been highlighted using "track changes" feature. Also, a native English speaker has been engaged to improve the fluency and the readability of the manuscript.

We believe that the revision proposed by the referees, and further implemented in the text, contributed to improve the manuscript. Thus, we kindly ask to and re-consider the manuscript for publication.

Please find a point-by-point response to the referees’ comments below.

Best regards,

Dr. Francesco Di Gennaro

Reviewer 4

The study is interesting and strategically important because it adopts a positive process based on information that should drive the most appropriate health planning. Such an approach is not easy and frequent in LMICs such as Mozambique. It must also be promoted by supporting this kind of study through a careful and proper peer review.

In conclusion, this study should be published after major revision  

 We thank you very much for the encouraging feedback on our manuscript.  We have considered all the useful suggestions you provided in the pdf.

Here the answers to your questions:

  1. Were they landline telephone and mobile phone? in latter case please describe how you collected the number

Response; The telephone number of participants were collected through the patients list of adolescent services SAAJs within the health centres, they were mainly mobile telephone. We better explained it within the test.

  1. Why school role as a source of information has not been investigated? Please explain

Response: thank you for having raised this important point we completely missed. We added this in the study limitations and we will surely consider it for any further similar studies.

General remarks.

It should be emphasized that this study is a preliminary descriptive one that will be developed by implementing other study designs focusing on the determinants you investigated so far.

It should be paid attention to the starting research hypothesis, which should be defined at the beginning of the article

We thank you very much for the encouraging feedback on our manuscript.  We have considered all the useful suggestions to improve our research and the two following statements were added:

Discussion section:“Further research, implementing other study designs focusing on the determinants emerged with our preliminary descriptive should be considered.”

Introduction: “The purpose of our research is therefore to study the awareness and understanding of SARS-CoV2 in young people, HIV positive, in two provinces in Mozambique: Sofala and Tete; the knowledge gap expected to be documented and identified could be helpful to inform target interventions.”

Round 2

Reviewer 1 Report

The extended introduction and dicussion now provide a better overview of the work.

Reviewer 4 Report

I've neither comments nor further suggestions. Thank you